# Ferroptosis Among the Antiproliferative Pathways Activated by a Lipophilic Ruthenium(III) Complex as a Candidate Drug for Triple-Negative Breast Cancer

**DOI:** 10.3390/pharmaceutics17070918

**Published:** 2025-07-16

**Authors:** Maria Grazia Ferraro, Federica Iazzetti, Marco Bocchetti, Claudia Riccardi, Daniela Montesarchio, Rita Santamaria, Gabriella Misso, Marialuisa Piccolo, Carlo Irace

**Affiliations:** 1Department of Molecular Medicine and Medical Biotechnology, School of Medicine and Surgery, University “Federico II” of Naples, 80131 Naples, Italy; mariagrazia.ferraro@unina.it; 2BioChem Lab, Department of Pharmacy, School of Medicine and Surgery, University “Federico II” of Naples, 80131 Naples, Italy; federica.iazzetti@unina.it (F.I.); marialuisa.piccolo@unina.it (M.P.); 3Department of Life Science, Health and Health Professions, Link Campus University, 00165 Rome, Italy; marco.bocchetti@unicampania.it; 4Department of Precision Medicine, University of Campania “Luigi Vanvitelli”, 80138 Naples, Italy; gabriella.misso@unicampania.it; 5Department of Chemical Sciences, University “Federico II” of Naples, 80126 Naples, Italy; claudia.riccardi@unina.it (C.R.); daniela.montesarchio@unina.it (D.M.)

**Keywords:** lipophilic Ru(III) complexes, antitumor Ru(III)-based drugs, triple-negative breast cancer (TNBC), iron, oxidative stress, regulated cell death (RCD), ferroptosis, preclinical cellular studies

## Abstract

**Background/Objectives**: In the context of preclinical studies, we have hitherto showcased that a low-molecular-weight ruthenium(III) complex we named AziRu holds significant potential for further developments as an anticancer candidate drug. When appropriately converted into stable nanomaterials and delivered into tumor cells, AziRu exhibits superior antiproliferative activity, benefiting from a multimodal mechanism of action. The activation of regulated cell death (RCD) pathways (i.e., apoptosis and autophagy) has been proved in metastatic phenotypes, including triple-negative breast cancer (TNBC) cells. This study focuses on a bioengineered lipophilic derivative of AziRu, named PalmiPyRu, that we are currently developing as a potential anticancer drug in preclinical studies. When delivered in this way, AziRu confirms a multimodal mechanism of action in effectively blocking the growth and proliferation of TNBC phenotypes. Special focus is reserved for the activation of the ferroptotic pathway as a consequence of redox imbalance and interference with iron homeostasis, as well as the glutathione biosynthetic pathway. **Methods**: Human preclinical models of specific TNBC phenotypes and healthy cell cultures of different histological origin were selected. After in vitro treatments, cellular responses were carefully analyzed, and targeted biochemical and molecular biology experiments coupled to confocal microscopy allowed us to explore the antiproliferative effects of PalmiPyRu. **Results**: In this study, we unveil that PalmiPyRu can enter TNBC cells and interfere with both the iron homeostasis and the cystine-glutamate antiporter system Xc-, causing significant oxidative stress and the accumulation of lipid oxidation products. The increase in intracellular reactive free iron and depletion of glutathione engender a lethal condition, driving cancer cells toward the activation of ferroptosis. **Conclusions**: Overall, these outcomes allow us, for the first time, to couple the antiproliferative effect of a ruthenium-based candidate drug with the inhibition of the Xc- antiporter system and Fenton chemistry, thereby branding PalmiPyRu as an effective multimodal inducer of ferroptosis. Molecular mechanisms of action deserve further investigations, and new studies are underway to uncover how interference with Xc- controls cell fate, allowing us to explore the connection between iron metabolism regulation, oxidative stress and RCD pathways activation.

## 1. Introduction

In the framework of preclinical trials, we demonstrated that AziRu, a low-molecular-weight ruthenium(III) complex (depicted in Figure 1), has the potential to be further developed as an anticancer drug candidate [1]. Indeed, when properly stabilized and delivered into cancer cells, it exhibits significant antiproliferative activity, benefitting from a multimodal mechanism of action that ultimately promotes the activation of different regulated cell death (RCD) pathways [2,3]. To achieve this goal, AziRu was originally converted into nanomaterials endowed with special features through decoration with highly functionalized nucleolipid- or amminoacyllipid-based scaffolds and subsequent co-formulation with biocompatible lipids [4,5,6,7]. The resulting stable liposomes have hitherto demonstrated superior antiproliferative efficacy in both in vitro and in vivo preclinical models of human tumors [1,8,9]. Of special interest were the outcomes achieved across different phenotypes of breast cancer (BC) [2,10]. Moving in this direction, we recently enriched the collection of Ru(III)-containing derivatives by designing bioengineered complexes featured by natural lipid moieties directly conjugated to the AziRu platform. Thus, a set of lipophilic Ru(III) complexes conceived as prodrugs was obtained as prospective candidate therapeutics. Their chemical behavior and biological properties were investigated in preliminary targeted bioscreens [11].

Once they have entered the cells, AziRu-functionalized nanosystems can reactivate tumor-suppressed RCD pathways, as the main mechanisms underlying their antiproliferative effect [3,8,9]. Nanoformulated AziRu has proved to be effective in the treatment of aggressive and metastatic tumor phenotypes, including triple-negative breast cancer (TNBC) [2,10]. Its multitarget action—shared with other potential Ru-based agents—is further supported by significant interference with cellular processes beyond RCD, such as those regulating the migration and invasion of metastatic malignant cells [2,12,13]. In this context, former studies on NAMI-A—a milestone in the development of bioactive ruthenium complexes—have already made assumptions about the antimetastatic potential of low-molecular-weight Ru(III) complexes [14]. In the meantime, BOLD-100, a metallotherapeutic structurally similar to AziRu, is currently in phase II clinical trials for the treatment of some advanced cancers, with favorable results [15]. Accordingly, a Ru(III)-based therapeutic has been proposed again in studies on humans, arousing new interest around anticancer Ru-based agents with evident efficacy [16,17]. Chemically, BOLD-100 represents the progress in iconic Ru(III) complexes, i.e., KP1339 and KP1019, all structurally related to AziRu and extensively characterized at the preclinical level [18,19,20]. Among the novel lipid-functionalized Ru(III) complexes, the palmitic acid-tailed compound named PalmiPyRu (depicted in Figure 1) has emerged as particularly effective in breast cancer cell (BCC) models, and preclinical studies revealed its favorable safety profile and biocompatibility in healthy cellular models [11]. Notably, in targeted bioscreens, PalmiPyRu exhibited a 20-fold enhanced cellular uptake compared to the naked AziRu. Overall, our findings suggest that PalmiPyRu is the best brand-new lipophilic candidate, worthy of further investigation and preclinical development [11].

Here, we demonstrate that PalmiPyRu can readily interact with biological targets in TNBC cell models and interfere with iron metabolism, causing a significant dysregulation in cellular homeostasis with a consequent induction of oxidative stress. In turn, oxidative stress can be concerned in iron-dependent RCD pathways, such as ferroptosis [21,22]. Cancer cells are known to differ markedly in their redox metabolism from healthy tissues [23,24]. Therefore, interference with cellular redox homeostasis seems an attractive and promising approach for cancer therapy [25]. In addition, targeting iron metabolism, frequently reprogrammed in tumor cells to support their growth and proliferation, equally remains an attractive option to block uncontrolled proliferation [26]. In this context, it is also interesting to highlight that chemical interferences between iron and ruthenium have been hypothesized. Both elements share some physico-chemical properties, including the potential to undergo redox processes and switch between several oxidation states under physiological conditions [27,28,29,30].

In addition to apoptosis and autophagy, here we show that ferroptosis is one of the RCDs detectable in TNBC preclinical models exposed to proper in vitro treatment with PalmiPyRu. This is in line with recent evidence indicating that different RCD mechanisms may share some pathways and have common inducers [31,32,33,34]. Indeed, new functions of Bcl-2 family proteins and their network with autophagy-related (ATG) proteins, including Beclin-1, in the crosstalk between mechanisms regulating cell death and survival are gradually emerging [35]. Since many tumors evade apoptosis, the simultaneous targeting of different RCD modalities becomes a promising option for effective antitumor therapies free from chemoresistance [36,37]. Therefore, using well-characterized preclinical models of TNBC, we have explored alterations in iron homeostasis induced by PalmiPyRu and their correlations with the endogenous production of reactive oxygen species (ROS). Oxidative damage to lipid biomolecules leading to an iron-dependent type of cell death has been examined. Overall, comprehensive cellular investigations and selective biomarkers analyses suggest the activation of ferroptosis among the possible RCD mechanisms underlying the antiproliferative effect of PalmiPyRu.

## 2. Experimental Section

### 2.1. Cancer and Healthy Cell Cultures

For this study, we selected a panel of triple-negative breast cancer (TNBC) cells endowed with definite phenotypes, as well as different types of healthy control cell lines.

### 2.2. Human TNBC Cell Lines

MDA-MB-231 cells were obtained from ATCC (HTB-26™), cultured in Dulbecco’s Modified Eagle Medium (DMEM) supplemented with 10% foetal bovine serum (FBS), 2 mM L-glutamine, 100 units/mL penicillin and 100 μg/mL streptomycin.

BT-549 cells were obtained from ATCC (HTB-122™), maintained in Roswell Park Memorial Institute (RPMI) medium supplemented with 10% FBS, 100 units/mL penicillin, 100 μg/mL streptomycin 100 μg/mL and 0.023 U/mL human insulin.

Hs 578T cells were kindly provided by Dr. De Palma Fatima at the Department of Molecular Medicine and Molecular Biotechnology, University Federico II of Naples, and were obtained from CEINGE–Biotecnologie Avanzate Franco Salvatore. Cells were cultured in DMEM supplemented with 10% FBS, 100 units/mL penicillin, 100 μg/mL streptomycin 100 μg/mL and 0.023 U/mL human insulin.

### 2.3. Human Healthy Control Cell Lines

HaCaT cells (Human Immortalized Keratinocytes) were kindly provided by Dr. Valeria Cicatiello at the “Italian National Research Council” (CNR), Institute of Genetics and Biophysics, Naples (Italy). Cells were grown in DMEM supplemented with 10% FBS, 100 units/mL penicillin and 100 μg/mL streptomycin.

MCF-10A cells (Non-Tumorigenic Mammary Epithelial Cells) were purchased from ATCC (CRL-10317™) and cultured in DMEM supplemented with 5% FBS, 500 ng/mL hydrocortisone, 100 ng/mL cholera toxin, 10 μg/mL insulin, 20 ng/mL epidermal growth factor (EGF), 100 units/mL penicillin and 100 μg/mL streptomycin.

Human Hair Follicular Keratinocytes (HHFKs) were obtained from ScienCell™ (Cat No. #2440) and maintained in keratinocyte medium (KM) enriched with 5 mL Keratinocyte Growth Supplement (KGS).

Primary Human Dermal Fibroblasts (HDFas) were purchased from ATCC (PCS-201-012™) and cultured in Fibroblast Basal Medium supplemented with Fibroblast Growth Kit–Low Serum.

All cells were grown in a humidified 5% carbon dioxide atmosphere at 37 °C and cultured according to manufacturer recommendations.

### 2.4. Cell Survival Assessment

The antiproliferative activity was evaluated by estimating a “cell survival index” through targeted bioscreens, which joins cell viability assessment (by the MTT functional assay) with automated cell count (quantitative assay by automatic cell counter). Specifically, the “cell survival index” is calculated as the arithmetic average between the percentage values derived from the MTT assay and automated cell counting [2,8,9]. For in vitro treatments, TNBC cell lines (MDA-MB-231, Hs 578T and BT-549) were seeded into 96-well culture plates at a density of 10^4^ cells/well and allowed to grow for 24 h. The culture medium was then replaced with fresh medium, and the cells were treated with PalmiPyRu at concentrations ranging from 5 to 150 μM for different times, depending on the type of experiment. Cisplatin (*c*DDP) was used as a reference drug (positive control).

The same experimental setup in vitro was applied to assess the biocompatibility of PalmiPyRu on healthy control cell lines (HDFas, HHFKs, HaCaTs and MCF7-10A). Cell viability was measured using the MTT assay (Sigma, Milan, Italy), which evaluates mitochondrial redox capacity to convert dissolved MTT into insoluble purple formazan. Briefly, after treatments, the medium was removed, and cells were incubated with 20 μL/well of MTT solution (5 mg/mL) for 1 h at 37 °C in a humidified 5% CO_2_ incubator. The incubation was stopped by removing the MTT solution and adding 100 μL/well of DMSO to solubilize the resulting formazan crystals. Finally, absorbance was measured at 550 nm using a microplate reader (ThermoFisher, Waltham, MA, USA). Total cell counting was determined using the TC20 automated cell counter (Bio-Rad, Milan, Italy), which provides accurate and reproducible total cell counts and a live/dead ratio in a single step using a specific dye exclusion assay (Trypan Blue). After treatment in vitro, the medium was removed, and cells were harvested. Then, a 10 μL aliquot of cell suspension was mixed with 0.4% Trypan Blue solution in a 1:1 ratio. The mixture was loaded into disposable slide chamber and analyzed. The instrument provided a total cell count (cells/mL), live cell count and percent viability, automatically adjusting for dilution if Trypan Blue was detected. The total counts and live/dead ratio from random samples for each cell line were subjected to comparisons with manual haemocytometers in control experiments.

### 2.5. IC_50_ Calculation

The calculation of the concentration required to inhibit the net increase in the cell number and viability by 50% (IC_50_) was based on plots of data (n = 5 for each experiment) and repeated three times (total n = 15). IC_50_ values were calculated from concentration–effect curves by nonlinear regression using a curve fitting program (GraphPad Prism 8.0) and are expressed as the mean values ± SEM (n = 15) of three independent experiments [2,8,9]. All subsequent in vitro experiments were performed using PalmiPyRu IC_50_ concentrations, calculated as previously described.

### 2.6. ROS Detection

Cellular ROS production was assessed using the ROS Detection Assay Kit (Canvax Biotech, Valladolid, Spain), based on dichlorodihydrofluorescein diacetate (H_2_DCFDA)—a fluorogenic dye that detects intracellular hydroxyl radicals, peroxides and other reactive oxygen species.

H_2_DCFDA is a cell-permeable non-fluorescent probe that passively diffuses into cells. Once inside, intracellular esterases cleave its acetyl groups, producing a non-fluorescent intermediate, which is subsequently oxidized by ROS into highly fluorescent 2′,7′-dichlorofluorescein (DCF). The fluorescence intensity is directly proportional to the intracellular ROS levels. Operatively, MDA-MB-231 and Hs 578T cells were seeded in a black 96-well plate at a density of 10^4^ cells/well. Cells were cultured at 37 °C with 5% CO_2_ for 24 h, reaching 70–80% confluence at the time of the experiment. Then, cells were incubated with H_2_DCFDA working solution (25 μM) at 37 °C for 30 min. H_2_DCFDA-containing solution was replaced with fresh medium containing an IC_50_ concentration of PalmiPyRu, FAC (5 μg/mL) or DFO (100 μM), respectively. Hydrogen peroxide (H_2_O_2_ 50 μM), a known ROS-inducing agent, was used as positive control. After 48 h of treatment, the fluorescence intensity was measured at Ex/Em = 485/530 nm using a fluorescence microplate reader (Promega, GloMax, Madison, WI, USA).

### 2.7. LIP Measurement

Intracellular iron concentration was assessed using FerroOrange (Dojindo, Munich, Germany), a fluorescent probe that selectively detects intracellular Fe^2+^in live cells [38]. MDA-MB-231 and Hs 578T cells were seeded into 96-well black plate at a density of 10^4^ cells/well and allowed to grow for 24 h. Then, cells were incubated with an IC_50_ concentration of PalmiPyRu for a further 48 h. Internal controls were tested to characterize the responsiveness toward different concentrations of intra-cellular iron. Specifically, ferric ammonium citrate (FAC, 5 μg/mL) was used as an iron source to increase iron levels; deferoxamine (DFO, 100 μM), was used as an iron chelator to reduce iron intracellular availability [39]. After a 48 h treatment, cells were incubated with 1 μM FerroOrange working solution at 37 °C for 30 min. The fluorescence intensity was measured with a fluorescent microplate reader (Promega, GloMax). In addition to the quantitative measurement of cellular iron content, cells were plated in four-well µ-Slide chambers at a concentration of 10^4^ cells per well and treated under the same experimental conditions described above, to monitor intracellular iron through microscopic observation. Cells were then examined using a Zeiss LSM 900 Airyscan 2 (Zeiss, Oberkochen, Germany) confocal microscope at 63× magnification (oil-immersion objective) for high-resolution visualization (Ex/Em: 540/580 nm).

### 2.8. Proteins Extraction

MDA-MB-231 cells were cultured in 60 mm culture dishes under standard conditions. Once cells reached sub-confluence, they were treated for 48 h with an IC_50_ concentration of PalmiPyRu, FAC (5 μg/mL), DFO (100 μM) or the ferroptosis inducer Erastin (10 µM), respectively. After treatments in vitro, cells were collected by scraping in phosphate-buffered saline (PBS) and centrifuged to obtain cellular pellets. Cells were then lysed at 4 °C for 30 min in RIPA buffer (25 mM Tris–HCl, pH 7.4, 150 mM NaCl, 5 mM EDTA, 5% (*v*/*v*) glycerol, 1% Triton) supplemented with protease inhibitors (TermoScientific). The protein fraction was collected by centrifugation at 13,000× *g* for 10 min at 4 °C. Concentration was determined using the Bio-Rad assay (Bio-Rad Laboratories, Milan, Italy) before proteins were stored at −80 °C [39].

### 2.9. Western Blot Analysis

Protein samples (30 μg) from MDA-MB-231 cell lysates were separated by SDS-PAGE and transferred onto nitrocellulose membranes. Membranes were then blocked at room temperature using milk buffer 5% (*w*/*v*). After 1 h, membranes were incubated overnight at 4 °C with the following primary antibodies: Transferrin Receptor 1 (TfR1–5 µg/mL, Invitrogen, Paisley, UK); Ferroportin-1 (SLC40A1–1:1000, Affinty Biotech, Houston, TX, USA); Ferritin Light Chain (1:1000, Affinty); GPX4 (1:1000, Affinty Biotech), ALOX15 (1:1000, Affinty Biotech 1:1000); SLC3A2 (1:1000, Affinty); SLC7A11 (1:1000, Affinty Biotech); Bcl-2 (1:1000, Affinity Biotech), Bax (1:500, Santa Cruz, Heidelberg, Germany); LC3B (1:1000, Cell Signaling, Danvers, MA, USA); and β-actin (1:2500, Invitrogen). Subsequently, membranes were incubated with appropriate secondary antibodies (anti-mouse or anti-rabbit, 1:10,000) for 1 h. The resulting immunocomplexes were detected by the ECL chemiluminescence method (ECL, Elabscience, Houston, TX, USA) and visualized using a ChemiDOC imaging system (Bio-Rad). Densitometric analysis was performed using ImageJ software, version 1.54p [39].

### 2.10. Glutathione (GSH/GSSG) Ratio Assay

The intracellular glutathione redox state was assessed using the GSH/GSSG Ratio Assay Kit (Abnova, Neihu, Taipei, Taiwan, Cat. No. KA6046). This fluorometric assay quantitatively determines the ratio of reduced glutathione (GSH) to oxidized glutathione (GSSG) in biological samples. Briefly, cells were seeded in six-well plates and treated with PalmiPyRu at its IC_50_ concentrations or Erastin (10 µM), used as a ferroptosis inducer. After treatment, cells were lysed through freeze–thaw cycles. Lysates were centrifuged at 13,000× *g* for 10 min at 4 °C, and the supernatant was collected for analysis. To prevent GSH oxidation, samples were deproteinized using 5% sulfosalicylic acid (SSA), followed by centrifugation at 8000× *g* for 10 min. A series of GSH and GSSG standard solutions were prepared by the serial dilution of the provided stock solutions in assay buffer to generate a standard curve. Then, 50 µL of either sample were loaded into a 96-well black fluorescence plate, and 50 µL of the reaction mixture was added to each well. The plate was incubated at room temperature for 30 min, protected from light. Fluorescence intensity was measured using a fluorescence microplate reader (Promega, GloMax) at Ex/Em 490/525 nm. The GSH and GSSG concentrations were determined by comparing the fluorescence intensity of samples to the standard curves.

### 2.11. Lipid Peroxidation

Lipid peroxide (LPO) levels in MDA-MB-231 cells were measured using the Lipid Peroxidation Assay Kit (E-BC-K176, Elabscience), which uses a C11-BODIPY 581/591 fluorescent probe for the detection of LPO production. Briefly, MDA-MB-231 cells were cultured in a 96-well microplate at a density of 10^4^ cells/well and allowed to grow until 70–80% confluence was reached. Then, cells were treated with the IC_50_ concentration of PalmiPyRu or Erastin (10 μM) for 48 h. After treatment, cells were incubated with a solution containing the fluorescent probe (100 μmol/L) at 37 °C for 30–60 min in the dark. Following incubation, the fluorescence intensity was measured at Ex/Em = 500/540 nm using a fluorescence plate reader (Promega, GloMax).

### 2.12. MDA Determination

Malondialdehyde (MDA), as an oxidative stress biomarker from lipid peroxidation, was determined by the MDA Assay Kit (abcam, ab118,970), which provides a colorimetric/fluorimetric highly specific tool for the sensitive detection of MDA in biological samples with little interference from other aldehydes. MDA can also indirectly reflect the level of cellular damage from oxygen free radicals. Briefly, MDA-MB-231 cells were cultured in 100 mm culture dishes at a density of 10^6^ cells/well and allowed to grow until 70–80% confluence was reached. Then, cells were treated with the IC_50_ concentration of PalmiPyRu or Erastin (10 μM) for 48 h. After treatment, cells were resuspended in 300 µL of MDA lysis buffer with 3 µL of the antioxidant BHT and homogenized on ice. Then, lysates were centrifuged at 13,000× *g* for 10 min. The resulting supernatants and MDA standards were each mixed with 600 µL of developer reaction mix and incubated at 95 °C for 60 min. After incubation, samples were cooled on ice, and 200 µL of each sample was transferred to a black 96-well microplate. Fluorescence was measured at Ex/Em = 532/553 nm using a fluorescence plate reader (Promega, GloMax).

### 2.13. Mitochondria Morphological Analysis by Confocal Microscopy

Morphological changes in the mitochondria of MDA-MB-231 cells after treatments in vitro were determined using the MitoTracker Red CMXRos selective probe (Ex\Em 579/599 nm). Briefly, 1.5 × 10^4^ cells were seeded in a four-well µ-Slide and incubated for 48 h. Then, cells were treated with the IC_50_ concentration of PalmiPyRu, *c*DDP (10 µM) or Erastin (10 µM), respectively, for a further 48 h. After incubations, cells were treated with FBS-free medium containing 100 nM MitoTracker at 37 °C for 20 min. To label the nuclei, a DAPI (Ex/Em 358/461 nm) solution (1:1000) was used at room temperature for 10 min of incubation. Fluorescence images were captured using a confocal microscope (Zeiss LSM 900 Airyscan 2) with a 63× magnification (oil-immersion objective).

### 2.14. RCD Pathway Activation

To evaluate the ability of PalmiPyRu to simultaneously trigger multiple RCD pathways, MDA-MB-231 cells were treated with the IC_50_ concentration of PalmiPyRu alone or in combination with one or more of the following specific inhibitors: Z-VAD-FMK (10 μM, an apoptosis inhibitor), Chloroquine (CQ) (10 μM, an autophagy inhibitor) and Ferrostatin-1 (2 μM, a ferroptosis inhibitor). After a 48 h treatment, cell survival was assessed as previously described.

### 2.15. Statistical Data Analysis

All experimental data were presented as mean values ± SEM. Statistical analysis was performed using one-way or two-way ANOVA followed by Dunnett’s or Bonferroni’s tests for multiple comparisons. GraphPad Prism 8.0 software was used for analysis. Differences between means were considered statistically significant when *p* ≤ 0.05 was achieved.

## 3. Results

### 3.1. PalmiPyRu: Evidence of Anticancer Efficacy and Selectivity In Vitro

The bioactivity of PalmiPyRu was first assessed in preclinical models of TNBC (MDA-MB-231, BT-549 and Hs 578T cells), which give rise to clones endowed with definite proliferative capacities, invasive potentials and resistance to therapy. The antiproliferative potential of PalmiPyRu was estimated by the “Cell survival index” analysis at different times and concentrations, as reported in the experimental section. Following treatments in vitro, at all the selected endpoints (24, 48 and 72 h), a significant concentration-dependent decrease in cell survival across all three cell lines was observed. As an illustrative example, Figure 2a shows evident cytotoxic effects after 48 h of PalmiPyRu application in the selected TNBC lines, with a significant concentration-dependent decrease in cell survival (data from other endpoints are provided in the Appendix A). In parallel, to assess the biocompatibility and selectivity of PalmiPyRu, we exposed a panel of healthy cell types to the same experimental conditions. Following treatments in vitro, PalmiPyRu demonstrated minimal cytotoxicity in normal mammary epithelial cells (MCF-10A), primary keratinocytes (HHFKs), primary dermal fibroblasts (HDFas), and immortalized keratinocytes (HaCaTs). Moderate interference with cell growth and proliferation was detected only in HaCaT cells at the highest tested concentration in vitro (150 µM) (Figure 2b). Interestingly, we observed similar cellular responses both in continuous and primary cultures, further endorsing PalmiPyRu biocompatibility in healthy cells. IC_50_ values, reported in Table 1, provide a quantitative comparison between PalmiPyRu and *c*DDP treatments in vitro. *c*DDP was used as reference drug and a positive control for cytotoxicity. Regarding TNBC cell lines, PalmiPyRu showed IC_50_ values in the low micromolar range, very close to those of *c*DDP and thereby indicative of significant antiproliferative activity. Noteworthily, PalmiPyRu confirmed significantly lower toxicity towards healthy cells than *c*DDP, suggesting a potential better safety profile compared to the reference drug. Overall, in preclinical bioscreens, PalmiPyRu demonstrated *c*DDP-like antiproliferative effects against TNBC cells while behaving in a biocompatible fashion, with a limited occurrence of side effects commonly associated with traditional chemotherapy in healthy cells.

### 3.2. PalmiPyRu Triggers Oxidative Stress and ROS Production in TNBC Models

To obtain an insight into the antiproliferative effect of PalmiPyRu, we evaluated by fluorescent analysis its ability to induce ROS production (specifically hydroxyl radicals and peroxides), which in turn can be closely related to cell death. Indeed, elevated cellular ROS levels promote oxidative stress, leading to DNA, protein, lipid and membrane damage. Moreover, alterations in cellular redox status can activate different RCD pathways, including ferroptosis. As shown in Figure 3, IC_50_ concentrations of PalmiPyRu significantly increased ROS production in a time-dependent manner. These biological effects were observed both in MDA-MB-231 (Figure 3a,b) and Hs 578T cells (Figure 3c,d). The net increase in ROS generation after in vitro incubation was very significant and time-dependent.

### 3.3. Disruption of Iron Homeostasis

Considering the interplay between the redox status regulation in tumor cells, iron chemistry in ROS generation and possible induction of cellular oxidative stress, we next examined the amount of labile iron pool (LIP) in the TNBC models (i.e., MDA-MB-231 and Hs 578T cells) exposed to PalmiPyRu. To this purpose, FerroOrange, a selective fluorescent probe for the detection of intracellular Fe^2+^, was used. Cells were loaded with ferric ammonium citrate (FAC) and deferoxamine (DFO) to ensure iron-repleted or iron-depleted conditions in vitro, respectively, as experimental controls for iron homeostasis reactivity to xenobiotic modulators [38]. Notably, we detected a significant increase in intracellular free iron (in the form of Fe^2+^) in both TNBC cell lines after a 48 h exposure to IC_50_ values of PalmiPyRu. This biological effect was very similar to that observed following FAC treatment in vitro (Figure 4a,b), suggesting an Ru-dependent interference with iron homeostasis, resulting in a significant increase in free reactive iron. Quantitative LIP analysis was supported by confocal microscopy bioimaging, confirming a significant increase in cellular free-iron levels (Figure 4c).

### 3.4. PalmiPyRu Modulates Key Proteins Involved in Iron Homeostasis in TNBC Model

Immunodetection analysis in MDA-MB-231 cells revealed an interesting PalmiPyRu-dependent modulation of some crucial proteins involved in iron homeostasis regulation. As shown Figure 5, Transferrin Receptor (TfR-1) expression was markedly downregulated by PalmiPyRu with respect to the untreated control, indicating that cellular iron uptake via TfR-1 is reduced, likely due to the LIP increase induced by the treatment. Following PalmiPyRu incubation in vitro, Ferroportin-1 expression was similarly reduced, suggesting decreased cellular iron export capacity. In parallel, intracellular ferritin levels were also markedly decreased. Considering ferritin’s role as the main cellular iron storage protein, this is consistent with the LIP increase we found after PalmiPyRu application in vitro [40,41]. FAC and DFO were used as controls for cellular iron homeostasis regulation, and the cellular responses observed under these conditions were consistent with expected patterns of iron homeostasis regulation [38].

### 3.5. PalmiPyRu Interferes with the Ferroptotic Protein Pathway

Based on the effects on iron cellular homeostasis and considering the pro-oxidative conditions observed in BC cells, we next investigated the expression of key proteins involved in ferroptosis. The aim was to provide insights into the potential role of iron in the activation of RCD pathways in response to PalmiPyRu treatment. In this frame, Erastin was used as positive control to induce the ferroptotic cell death by inhibiting the cystine-glutamate antiporter system Xc^−^. Cells treated with Erastin are deprived of cysteine and are unable to synthesize the antioxidant glutathione (GSH) [20]. Notably, the in vitro PalmiPyRu treatment of MDA-MB-231 cells caused a significant reduction in the expression of the Xc^−^ system, equally affecting both its heavy (SLC3A2) and light chain (SLC7A11). The extent of this downregulation was very similar to that induced by Erastin (see blots in Figure 6) [42]. Glutathione peroxidase 4 (GPX4) immunodetection analysis showed a nearly complete disappearance of this key enzyme engaged in cell protection from oxidative stress, suggesting cells and particularly membranes as potentially exposed to oxidative damage [43]. Under the same experimental conditions, we found a significant increase in the expression of arachidonate 15-lipoxygenase 1 (ALOX15) after PalmiPyRu treatment. LOXs are iron-dependent enzymes involved in lipid peroxide formation, and their overexpression and activity are strictly correlated with ferroptosis induction [44]. Therefore, the treatment of MDA-MB-231 cells with PalmiPyRu seems to promote the production of cellular lipoperoxides while inhibiting the possibility of their detoxification. Moreover, ALOX15 activity is accepted to be enhanced by high levels of intracellular iron and ROS, both conditions occurring in this experimental model [45]. Collectively, these findings suggest that PalmiPyRu predisposes TNBC cells to ferroptotic cell death. LC3B, i.e., the microtubule-associated protein 1 light chain 3B (MAP1LC3B), is engaged in autophagy regulation as well as in other critical biological processes such as apoptosis and differentiation. It is deemed the mammalian homolog of the autophagy-related protein 8 (ATG8). In addition, its involvement in ferroptosis has recently been assumed [46]. As far as LC3B expression is concerned, the blots in Figure 6 show increased levels of both isoforms (LC3B-I and LC3B-II) of this protein following PalmiPyRu treatment. Interestingly, LC3 activation could be also linked to ferritinophagy, responsible for ferritin degradation throughout autophagic pathways [46]. The downregulation of Bcl-2 and upregulation of Bax after PalmiPyRu treatment confirm the activation of the apoptotic machinery. These findings are in line with our former studies demonstrating that AziRu-containing nanoformulations can induce multiple RCD pathways, including autophagy and apoptosis [9].

### 3.6. Glutathione (GSH) Depletion and Lipid Peroxidation

To confirm that interferences induced by PalmiPyRu with proteins of the ferroptotic pathway can impact cellular redox homeostasis and increase vulnerability to ferroptosis, we next investigated the intracellular concentration of GSH as well as lipoperoxide generation in MDA-MB-231 cells. As depicted in Figure 7a by a fluorimetric assay, PalmiPyRu treatment induced a significant reduction of GSH, endorsed by a worthy increase in its oxidized form (GSSG). In addition, we found a substantial increase in lipid peroxides (Figure 7b), which is consistent with the reduction of GPX4 expression and the increase in ALOX15. The fluorimetric detection of MDA as a lipid peroxidation marker from polyunsaturated fatty acids confirms this outcome (Figure 7c). Overall, these effects closely resemble those observed with Erastin, used under the same experimental conditions, and further confirm the ability of PalmiPyRu to predispose TNBC cells to ferroptosis.

### 3.7. PalmiPyRu Effect on Mitochondria in MDA-MB-231 Cells

Considering mitochondria as central players in different programmed cell death pathways, we next assessed the ability of PalmiPyRu to direct MDA-MB-231 cells towards ferroptosis by exploring mitochondrial alterations through confocal microscopy. To this aim, mitochondria were labelled with MitoTracker (Figure 8). The experimental design envisioned the use of Erastin and *c*DDP as reference drugs to induce ferroptosis and apoptosis, respectively. Following treatment in vitro with *c*DDP, the red fluorescent signal was significantly reduced compared to untreated cells. Specifically, mitochondria appeared damaged and mainly located in the perinuclear region, suggestive of mitochondrial network disintegration, which is considered a canonical hallmark of apoptosis [47]. Erastin application to cells also lowered the mitochondrial fluorescent signal with respect to control cells, but less evidently than *c*DDP. However, in line with recent findings, morphological changes and cellular distribution dynamics concerning mitochondria also remained evident after Erastin treatment compared to controls [48]. Remarkably, MDA-MB-231 cells incubated with PalmiPyRu showed equally significant changes in mitochondrial structure and distribution, but the observed alterations could be defined as “in-between” compared to those observed after the single treatments with *c*DDP and Erastin. Overall, throughout the pattern of PalmiPyRu-induced biological effects, mitochondria were shrinking and became small, assuming features that are generally detected after ferroptosis activation.

### 3.8. PalmiPyRu Triggers Multiple RCD Pathways in MDA-MB-231 Cells

Our previous studies identified the nanoformulated AziRu complex as a multitarget agent able to induce multiple cell death pathways, including apoptosis and sustained autophagy. Hence, the overexpression of key proteins, such as Bax and beclin-1, orchestrating RCD processes has been highlighted previously [1,8,9,10]. Increased expression of the autophagic marker LC3 and the appearance of mitochondrial hallmarks of RCDs after PalmiPyRu treatment further support the activation of autophagy and apoptosis, even under ferroptotic conditions. To assess the coexistence of autophagic, apoptotic and ferroptotic mechanisms, we next performed targeted experiments by means of specific inhibitors to selectively block RCD processes, i.e., Z-VAD-FMK for apoptosis, Chloroquine for autophagy and Ferrostatin-1 for ferroptosis, respectively. As shown in Figure 9, the association of PalmiPyRu with each of these inhibitors led to a significant increase in the “Cell survival index” of TNBC cells compared to PalmiPyRu treatment alone. Interestingly, the contribution of each inhibitor in the recovery of cell survival seems to be roughly the same, suggesting that apoptosis, autophagy and ferroptosis are synergistic players connected to the antiproliferative effect of PalmiPyRu. Accordingly, PalmiPyRu’s antitumor effect almost completely disappeared when all three inhibitors were concurrently administered to MDA-MB-231 cells. This outcome proves that PalmiPyRu antiproliferative effect in TNBC cells is driven by the simultaneous activation of multiple RCD pathways.

## 4. Discussion

TNBC encompasses a very heterogeneous group of cancers among the most challenging BC types to treat [49]. TN phenotypes are also increasingly diagnosed in young women and represent a very aggressive and metastatic BC variant for which we still have no effective treatments [50,51]. New therapeutic tools are thereby required, and multitarget metal-based candidate drugs designed as non-platinum transition metals represent an approach that may prove to be effective [52,53,54,55]. As proof of this, a therapeutically developed Ru(III) complex called BOLD-100 recently reached human trials, re-proposing the option of evaluating an Ru-based candidate drug at the clinical level [15,16,17,18]. Scientists from both academia and industry have been focusing for a long time on the development of Ru-based complexes with evident bioactivity, which are potentially less toxic than conventional metallotherapeutics [16]. To a lesser extent reactive in biological environments than Ru(II) but therefore also less dangerous, Ru(III)-based agents possess several interesting properties from a medicinal point of view, supporting the design and development of several different derivatives. They can participate in many biological redox reactions and interact with different potential cellular targets by specific molecular interactions, upholding the multimodal mechanism of action that this type of drug candidate has hitherto demonstrated [10,12,16].

Herein, we showcase the antiproliferative effects in TNBC in vitro models of PalmiPyRu, a palmitoyl-conjugated Ru(III) complex designed as a lipophilic prodrug of the low-molecular-weight AziRu platform. In recent times, we demonstrated that, when properly delivered into cancer cells (e.g., functionalized with nucleolipid-based scaffold and then loaded in biocompatible liposome formulations), AziRu is capable of activating multiple RCD pathways in BC cells, including apoptosis and autophagy [2,3,4,6,7,8,9,10,56]. Now, we reveal PalmiPyRu’s ability to also trigger ferroptosis in TNBC cells, thus confirming its multimodal mechanism of action. From this perspective, PalmiPyRu can be framed as both a class I and IV ferroptosis inducer (FIN). Indeed, PalmiPyRu downregulates the system Xc^−^ antiporter, impairing cystine uptake and consequently reducing intracellular GSH levels. Tumors typically can benefit from elevated antioxidant systems to counteract oxidative stress [24,57,58]. More than others, the GSH system is crucial for the activity of the most effective cellular antioxidant enzymes [59,60]. Our findings show a significantly downregulated expression of the heterodimeric cystine-glutamate antiporter system Xc^−^. Two subunits are implied in the activity of this reverse transporter, i.e., the light chain xCT (SLC7A11) engaged in the actual antiporter activity, and the heavy chain 4F2hc (SLC3A2) regulating the cellular trafficking of Xc^−^ [42,61]. The action of PalmiPyRu appears to involve both subunits equally. Since cellular cysteine availability is believed to be a rate-limiting step in GSH synthesis, the ensuing biological effect is a substantial decrease in GSH formation [42,60]. Accordingly, we observed a significant decrease in cellular GSH levels. Overall, interference with Xc^−^ system expression at the cytoplasmic membrane allows PalmiPyRu to be considered a class I ferroptosis inducer. Its biological effect is very similar to that of the ferroptosis inducer Erastin, well-known for its ability to functionally inhibit the antiporter system Xc^−^ [42].

Not by chance, system Xc^−^ is frequently upregulated in cancer, where it is associated with advanced tumor stage and poor overall survival, providing tumor cells with a special antioxidant ability [23,62]. System Xc^−^ and the GSH pathway are thereby deemed attractive targets for anticancer therapies [63]. Moreover, by increasing GSH levels, system Xc^−^ activity is also correlated to resistance onset in many types of chemotherapeutic regimens [63,64]. Thus, as well as being an effective oncotherapeutic target, system Xc^−^ can be envisioned as further druggable to prevent the development of chemoresistance [63]. Since the discovery of Erastin in 2003, different molecules capable of specifically inhibiting the Xc^−^ antiporter have been identified [42]. Although Erastin itself has shown potential for cancer therapy, many limitations in its possible therapeutic use have emerged, prompting the search for appropriately functionalized analogues [65,66]. Moreover, recent findings provided evidence for pathological changes in healthy tissues after in vivo administration in mice, suggestive of potential Erastin-dependent systemic toxicity [67]. Candidate drugs among FINs I also include other system Xc^−^ inhibitors, such as the FDA-approved sulfasalazine (SSZ) and sorafenib. Both, however, have shown just as many limitations for further developments, thereby driving research towards the identification of new effective lead compounds for Xc^−^ inhibition [68,69,70]. Concerning the therapeutic approach by acting as a class IV FIN, we have detected that PalmiPyRu application in TNBC cells is effectively associated with interference with iron-related proteins and iron metabolism. Exploitation of the iron biological properties is fundamental for the growth and proliferation of many tumors that consequently deregulate iron metabolism for their own purposes [21,57]. This makes proteins concerned in iron metabolism equally attractive druggable targets in the framework of antitumor therapies [26]. Specifically, preclinical evidence shows significant variation in the expression of proteins engaged in the maintenance of iron homeostasis. PalmiPyRu’s biological effect somewhat resembles the iron overload condition, causing a significant increase in LIP levels correlated to a marked condition of iron-dependent oxidative stress [39,70]. Indeed, this treatment results in a substantial decrease in proteins that control iron homeostasis, including ferritin intracellular levels and TfR expression at the cytoplasmic membrane. We suppose the downregulation of ferritin—a cytoprotective protein with a powerful antioxidant and iron storage capacity—is critical in this context and probably associated with ferritinophagy, a selective autophagic degradation process at the interface between ferroptosis and autophagy [40,41,71]. LC3 activation after PalmiPyRu treatment endorses this hypothesis. In further support, we already showed the activation of sustained autophagy in BC cells after in vitro treatment with AziRu-loaded nanosystems [9]. Ferritinophagy implies the selective lysosomal degradation of ferritin mediated by LC3 and occurs in some autophagic pathways. It is currently considered a process at the interface between ferroptosis and autophagy that can play a significant role in cancer [71,72]. The subsequent iron release can impact cell fate differently, ranging from favoring tumor growth to being associated with the Fenton reaction and redox lipid death [21,57,71,72]. In response to treatment with PalmiPyRu, we found a considerable increase in intracellular free iron, which suggests a disruption of iron homeostasis in TNBC.

Moreover, targeted experiments after PalmiPyRu application supported large iron-dependent ROS production causing broad oxidative damage, likely directing TNBC cells towards ferroptosis. Accordingly, important lipid oxidative degradation was detected. Massive ROS production can damage lipids within cell membranes, specifically polyunsaturated fatty acids (PUFAs) more vulnerable to oxidative stress, causing the formation of lipid radicals collectively referred to as lipid peroxides or lipid oxidation products (LOPs) [43,73,74]. Accordingly, we also found an increase in the expression of enzymes involved in the peroxidation of PUFAs. Thus, from this perspective, PalmiPyRu can be deemed as a class IV FIN capable of triggering iron-dependent oxidative stress. Additionally, a possible direct interaction of PalmiPyRu with tumor cell lipids cannot be excluded, which could cause LOPs production triggered by redox processes involving ruthenium oxidation state switch. Oxidative stress induced by PalmiPyRu in TNBC cells may represent the link between the activation of ferroptosis and other RCD pathways. In fact, ROS have been associated with both the modulation of Bcl-2 proteins and mitochondrial disfunction, causing the sequential activation of the proapoptotic Bax-dependent pathway [75,76]. Accordingly, we have previously documented the mitochondrial proapoptotic effects of AziRu when appropriately nanodelivered in mammary cancers, which culminate with Bax upregulation and caspase activation [8]. In support, we have now observed canonical hallmarks of apoptosis overlapping with those of ferroptosis. Just recently, ferroptosis in the presence of apoptosis has been reported in human preclinical models of tumors, and emerging evidence proves that different RCDs can share similar activation pathways [31,32,33,34]. Additionally, more and more connections are gradually emerging between Bcl-2 proteins as master regulators of apoptosis and Atg-related proteins (e.g., Beclin-1) as central players in autophagy [35,77]. From this perspective, different cell death mechanisms could be prompted by the same inducers—including oxidative stress and ROS—thereby admitting the possible coexistence of different RCD pathways to describe drug-dependent antiproliferative effects [33,34,78].

## 5. Conclusions

We have demonstrated that the Ru(III) complex AziRu, bioengineered into a specific lipophilic palmitoyl-based derivative (i.e., PalmiPyRu), exhibits an anticancer mechanism based on the activation of multiple RCD pathways, including ferroptosis. This multimodal biological activity makes PalmiPyRu effective as a potential candidate drug, even in resistant tumors with limited therapeutic options such as TNBC.

To the best of our knowledge, this is the first time the antiproliferative effect of a ruthenium complex has been associated with the inhibition of the Xc^−^ antiporter. The ability to concurrently act by promoting iron-dependent oxidative stress and inhibiting the Xc^−^ antiporter brands PalmiPyRu an effective multimodal inducer of ferroptosis, endowed with the typical multitarget action of many ruthenium-based drug candidates. Molecular mechanisms of action deserve further investigations, and new studies are underway to uncover the connection between redox imbalance, iron metabolism and RCD pathway activation. Multi-targeted metallotherapeutics might yield new opportunities in the treatment of cancers that involve manifold pathogenic factors. Our findings further strengthen these concepts, proving that “mixed” RCD pathways may occur under specific conditions, such as those triggered by a specific ruthenotherapy in vitro. A feasible assumption is that oxidative stress and impaired iron homeostasis may be among the common inducers of different RCD pathways activation in response to PalmiPyRu treatment.

## Figures and Tables

**Figure 1 pharmaceutics-17-00918-f001:**
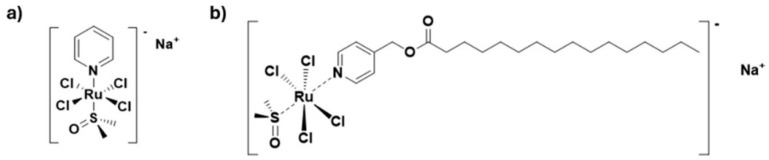
Molecular structures of Ru(III)-based complexes. (**a**) The low-molecular-weight Ru(III) complex AziRu, inspired by NAMI-A. (**b**) The novel lipid-functionalized Ru(III) complex named PalmiPyRu—a bioengineered palmitic acid-tailed compound.

**Figure 2 pharmaceutics-17-00918-f002:**
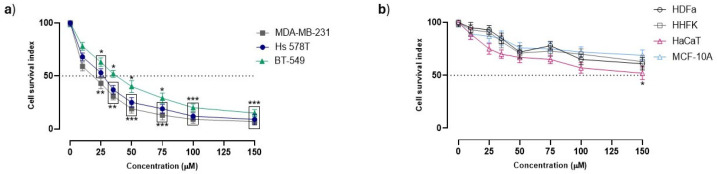
Antiproliferative effects of PalmiPyRu in TNBC cells and healthy cultures. Cell survival index, evaluated by the MTT assay and live/dead cell ratio analysis, for (**a**) TNBC cell lines (MDA-MB-231, Hs 578T, BT-549) and (**b**) non-cancerous cell models (MCF-10A, HHFKs, HDFas, HaCaTs), as indicated in legends, after a 48 h treatment with different concentrations of PalmiPyRu (range 5→150 μM). Data are plotted in line graphs as percentage of untreated control cells and are reported as mean of three independent experiments ± SEM (*n* = 15). * *p* < 0.05 with respect to control cells; ** *p* < 0.01 with respect to control cells; *** *p* < 0.001 with respect to control cells.

**Figure 3 pharmaceutics-17-00918-f003:**
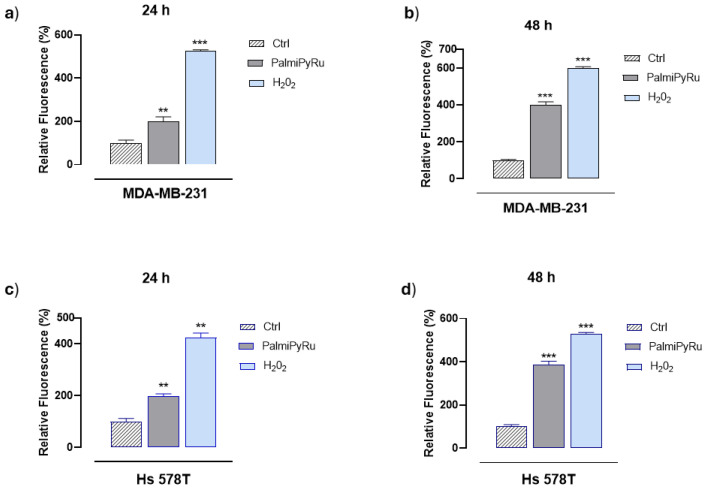
ROS generation in TNBC cells following PalmiPyRu treatment in vitro. Intracellular ROS levels in MDA-MB-231 (**a**,**b**) and Hs 578T (**c**,**d**) cells evaluated by fluorescent analysis after 24 h and 48 h of incubation with IC_50_ concentrations of PalmiPyRu (18 and 24 µM, respectively). H_2_O_2_-treated cells were used as positive controls for ROS generation. Results are plotted in bar graphs as relative fluorescence units and are the mean of three independent experiments ± SEM (*n* = 15). ** *p* < 0.01 with respect to control cells; *** *p* < 0.001 with respect to control cells.

**Figure 4 pharmaceutics-17-00918-f004:**
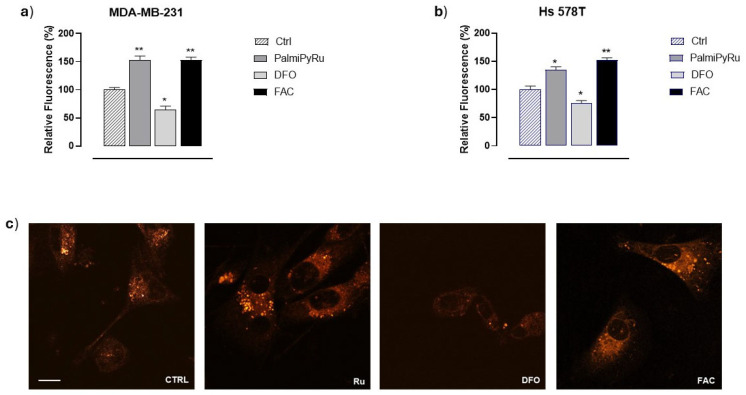
LIP levels in TNBC cells after PalmiPyRu treatment. Fluorescent analysis indicative of LIP levels in MDA-MB-231 cells (**a**) and Hs578T cells (**b**) after a 48 h treatment with IC_50_ concentrations of PalmiPyRu (18 and 24 µM, respectively). Deferoxamine (DFO, 100 µM) and ferric ammonium citrate (FAC, 5 µg/mL), as an iron chelator and an iron source, respectively, were used as internal controls to ensure iron depleted and iron repleted conditions. Data are plotted in bar graphs as relative fluorescence units and are the mean of three independent experiments ± SEM (*n* = 15). * *p* < 0.05 with respect to control cells; ** *p* < 0.01 with respect to control cells. (**c**) Representative images by confocal microscopy of MDA-MB-231 cells treated for 48 h with the IC_50_ concentration of PalmiPyRu (Ru), deferoxamine (DFO) or ferric ammonium citrate (FAC), respectively. FerroOrange (1 µM) was used as selective fluorescent probe for intracellular Fe^2+^ detection (Ex/Em: 540/580 nm). The scale bar represents 25 µm. The shown microphotographs are acquired at 63× magnification (oil-immersion objective).

**Figure 5 pharmaceutics-17-00918-f005:**
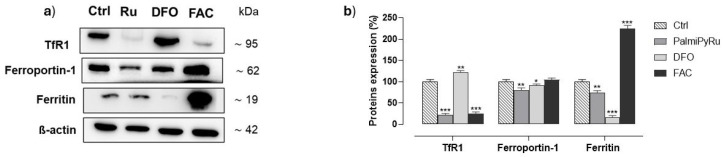
Expression of iron-related proteins in TNBC cells. (**a**) Representative blots showing the effect of the IC_50_ concentration (18 µM) treatment of PalmiPyRu (Ru) following 48 h of incubation in MDA-MB-231 cells on TfR1, Ferroportin-1 and ferritin expression. Treatments in vitro with DFO (100 µM) and FAC (5 µg/mL) were used as internal controls to resemble iron depleted and iron repleted conditions, respectively. The shown blots are representative of three independent experiments and are cropped from different parts of the same gel, as explicit by using clear delineation with dividing lines and white space. (**b**) After chemoluminescence, the immune complexes were quantified by densitometric analysis and plotted in bar graphs as percentage of controls. Shown are the average ± SEM values of three independent experiments. The anti-β-actin antibody was used to standardize the amount of proteins in each lane. * *p* < 0.05 with respect to control cells; ** *p* < 0.01 with respect to control cells; *** *p* < 0.001 with respect to control cells.

**Figure 6 pharmaceutics-17-00918-f006:**
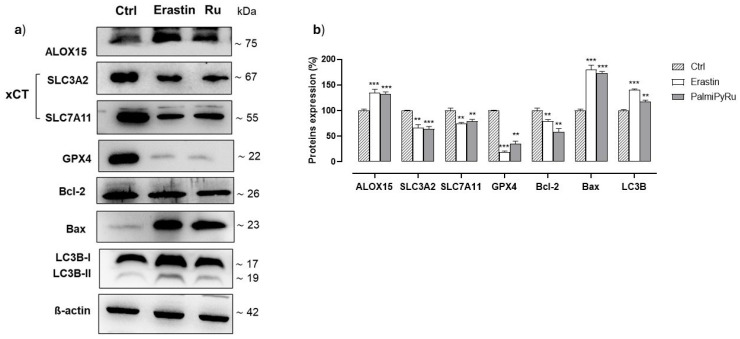
Immunodetection of proteins involved in RCD pathway. (**a**) Representative blots showing the effect of the IC_50_ concentration (18 µM) treatment of PalmiPyRu (Ru) or Erastin (10 µM) in MDA-MB-231 cells after 48 h of incubation on the expression of ALOX15, SLC3A2, SLC7A11, GPX4, Bcl-2, Bax and LC3B (LC3B-I and LC3B-II). The shown blots are representative of three independent experiments and are cropped from different parts of the same gel, as explicit by using clear delineation with dividing lines and white space. (**b**) After chemoluminescence, immunocomplexes were quantified by ImageJ software and plotted in bar graphs as percentage of controls. The average ± SEM values of three independent experiments are here shown. The anti-β-actin antibody was used to standardize the amount of proteins in each lane. ** *p* < 0.01 with respect to control cells; *** *p* < 0.001 with respect to control cells.

**Figure 7 pharmaceutics-17-00918-f007:**
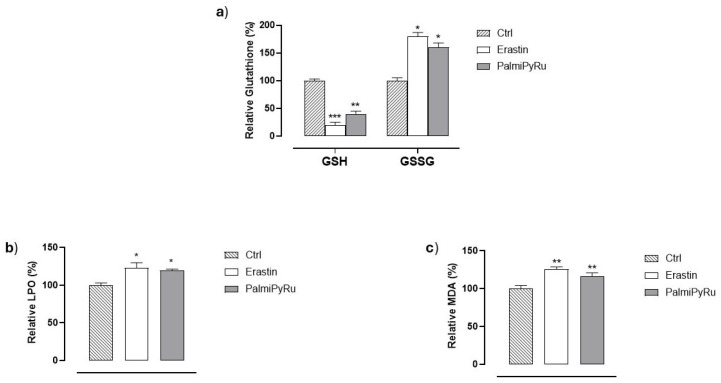
Glutathione (GSH) levels and lipoperoxidation (LPO) analysis. (**a**) Quantification by fluorimetric assay of reduced (GSH) and oxidized (GSSG) glutathione in MDA-MB-231 cells after 48 h of treatment in vitro with PalmiPyRu (18 µM) or Erastin (10 µM, used as ferroptosis inducer). (**b**) Lipid peroxidation (LPO) levels by a specific fluorescent probe in MDA-MB-231 cells following 48 h of incubation with PalmiPyRu (18 µM) or Erastin (10 µM). (**c**) Malondialdehyde (MDA) generation by a selective fluorimetric assay in MDA-MB-231 TNBC cells after a 48 h treatment with PalmiPyRu (18 µM) or Erastin (10 µM). Data are plotted in bar graphs as percentage of untreated control cells and are the average ± SEM values of three independent experiments (*n* = 15). * *p* < 0.05 with respect to control cells; ** *p* < 0.01 with respect to control cells; *** *p* < 0.001 with respect to control cells.

**Figure 8 pharmaceutics-17-00918-f008:**
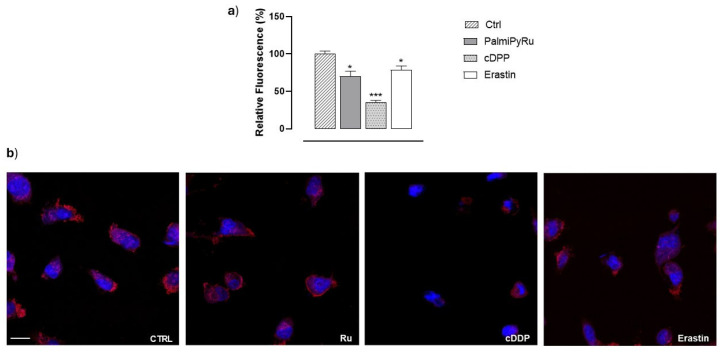
Mitochondrial analysis by confocal microscopy. (**a**) Quantification of the red fluorescence (Ex\Em 579/599 nm) by the use of the selective probe MitoTracker after application for 48 h of the IC_50_ concentration of PalmiPyRu (18 µM), *c*DDP (10 µM) or Erastin (10 µM), respectively, in MDA-MB-231 cells. Relative fluorescence is plotted in a bar graph as percentage of untreated control cells and is the average ± SEM values of three independent experiments. * *p* < 0.05 with respect to control cells; *** *p* < 0.001 with respect to control cells. (**b**) Representative confocal microphotographs following treatments in vitro for 48 h with PalmiPyRu (Ru), *c*DDP or Erastin, respectively, in MDA-MB-231 cells. Mitochondria were labelled with MitoTracker Red (Ex\Em 579/599 nm) and nuclei with DAPI (Ex/Em 358/461 nm), as indicated in the experimental section. The scale bar represents 15 µm. Fluorescence images were captured by Zeiss LSM 900 Airyscan 2 confocal microscope with a 40× magnification (oil-immersion objective).

**Figure 9 pharmaceutics-17-00918-f009:**
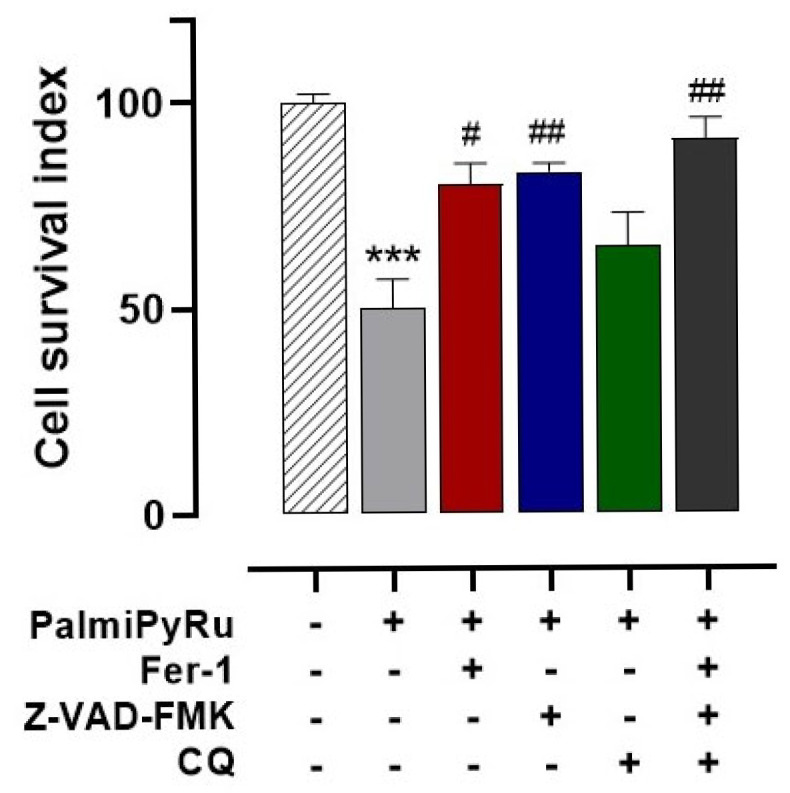
PalmiPyRu induces multiple RCDs pathways activation. Cell survival index was evaluated after 48 h treatments of MDA-MB-231 cells with PalmiPyRu alone (at its IC_50_ concentration, i.e., 18 µM) or in combination with selective RCD inhibitors, i.e., Ferrostatin-1 (2 μM, a ferroptosis inhibitor), Z-VAD-FMK (10 μM, an apoptosis inhibitor), Chloroquine (CQ 10 μM, an autophagy inhibitor), as indicated in the legend. Data are plotted in a bar graph as percentage of untreated control cells and are reported as the average of three independent experiments ± SEM (*n* = 15). *** *p* < 0.001 with respect to control cells; ^#^ *p* < 0.05 with respect to PalmiPyRu-treated cells; ^##^ *p* < 0.01 with respect to PalmiPyRu-treated cells.

**Table 1 pharmaceutics-17-00918-t001:** IC_50_ values (µM) of PalmiPyRu and cisplatin (*c*DDP), used in the same experimental condition as reference cytotoxic drug, in the indicated TNBC and healthy cell lined following 48 h of incubation in vitro. IC_50_ values were calculated from concentration–effect curves by nonlinear regression using GraphPad Prism 8.0 and are expressed as mean values ± SEM (*n* = 15) of three independent experiments.

IC_50_ Values (µM)							
Drugs	TNBC Cell Lines			Healthy Cell Lines			
	Hs578T	MDA-MB-231	BT-549	HDFa	HHFK	HaCaT	MCF-10A
cDDP	20 ± 4	11 ± 3	15 ± 2	72 ± 4	93 ± 3	50 ± 3	88 ± 3
PalmiPyRu	24 ± 3	18 ± 2	37 ± 5	>150	>150	>150	>150

## Data Availability

The original contributions presented in this study are included in the article. Further inquiries can be directed to the corresponding authors.

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
