# Peer review of "Ferroptosis Among the Antiproliferative Pathways Activated by a Lipophilic Ruthenium(III) Complex as a Candidate Drug for Triple-Negative Breast Cancer"

_pharmaceutics, 2025, doi:10.3390/pharmaceutics17070918_

Round 1

Reviewer 1 Report

Comments and Suggestions for Authors

The article by Maria Grazia Ferraro et al. «Ferroptosis among the antiproliferative pathways activated by a lipophilic ruthenium(III) complex as a candidate drug for triple negative breast cancer» is devoted to complex studies of the antiproliferative effect of the lipophilic derivative of ruthenium(III) complex named PalmiPyRu and to the study of possible mechanisms of RCD. The article may be published in the journal Pharmaceutics after minor revisions. It is necessary to present the AziRu and PalmiPyRu compounds in graphic form in the text of the article, and not only in the graphic abstract Take into account not only the articles cited by the authors, but also other works, for example, https://doi.org/10.1039/D2NR02994D (ruthenium complexes from monotherapy to combination therapy) https://doi.org/10.1017/S143192761300175X (Ru compound distribution within the cell MDAMB231, the mechanism of its activity, and its cellular targets.) If the authors want to use the file pharmaceutics-3737239-original-images.pdf as additional materials, in this case it is necessary to edit it. Number the figures with their description, improve their quality and add the figure numbers to the main text of the article.

Author Response

We appreciate the positive feedback and valuable suggestions from reviewer#1. We have thus further improved the manuscript as follows:

1.It is necessary to present the AziRu and PalmiPyRu compounds in graphic form in the text of the article, and not only in the graphic abstract .

1. Done. We have added a new figure (Fig. 1) concerning the molecular structures of AziRu and PalmiPyRu to the revised version of the paper.

2. Take into account not only the articles cited by the authors, but also other works, for example, https://doi.org/10.1039/D2NR02994D (ruthenium complexes from monotherapy to combination therapy) https://doi.org/10.1017/S143192761300175X (Ru compound distribution within the cell MDAMB231, the mechanism of its activity, and its cellular targets.)

2. The suggested references have been included in the revised paper.

3. Number the figures with their description, improve their quality and add the figure numbers to the main text of the article.

3. This has also been done. All figures have been inserted as high-resolution tiff files (400 dpi).

Reviewer 2 Report

Comments and Suggestions for Authors

This manuscript can be accepted after major revision.

How exactly does PalmiPyRu disrupt iron homeostasis? Are there specific molecular targets involved?

To ensure compliance with academic standards, authors must attach great importance to formatting specifications. It is essential to meticulously structure the entire manuscript in strict adherence to Pharmaceutics's formatting guidelines, leaving no detail overlooked.

How does the lipophilic modification enhance cellular uptake and tumor targeting compared to the parent compound AziRu? Does this modification improve membrane permeability or enable specific interactions with tumor cell lipids?

In TNBC cells, does PalmiPyRu primarily induce ferroptosis, or does it synergize with other RCD pathways? Are there experimental data showing combinatorial effects on cell death pathways?

Which TNBC preclinical models were used to validate PalmiPyRu’s activity? What were the key endpoints measured?

What are the key advantages of PalmiPyRu over AziRu in terms of efficacy, stability, or delivery?

This is a research article, and the author need not cite an excessive number of references.

Author Response

We thank reviewer #2 for his positive feedback and for all the observations that allowed to increase the quality of our paper.

1. How exactly does PalmiPyRu disrupt iron homeostasis? Are there specific molecular targets involved?

1. This is a very interesting aspect probably related to the multimodal action of AziRu and more generally of ruthenium(III) complexes. We found that our experimental data on interference with iron metabolism are substantially in agreement with those of other authors who have shown an elevation of iron levels in TNBC cells related to alterations of mitochondrial bioenergetics, redox balance and ROS production (https://doi.org/10.1016/j.jinorgbio.2021.111380). Energy balance and redox homeostasis are evidently oncological targets of primary importance given the sensitivity of cancer cells that must balance proliferation with cell survival. However, no hypotheses have been hitherto made on possible cellular targets that can cause alterations of iron homeostasis. What we can currently hypothesize, as reported in the paper, is an activation of pathways such as autophagy leading to ferritinophagy as key factor implicated in the elevation of iron levels. Despite the demands for iron, this goes beyond the capacities of tumor cell to counteract iron elevation. However, we have recently undertaken other studies based on both proteomic and computational approaches that could shed light on the actual biomolecular targets of AziRu underlying biological effects we observe in cancer cells, including those on iron metabolism. We expect to obtain interesting results in the immediate future.

2. To ensure compliance with academic standards, authors must attach great importance to formatting specifications. It is essential to meticulously structure the entire manuscript in strict adherence to Pharmaceutics's formatting guidelines, leaving no detail overlooked.

2. As suggested, we have been careful about the guidelines of “Pharmaceutics”.

3. How does the lipophilic modification enhance cellular uptake and tumor targeting compared to the parent compound AziRu? Does this modification improve membrane permeability or enable specific interactions with tumor cell lipids?

3. As reported in the introduction of the paper, the conversion of AziRu into a lipophilic derivative, i.e., PalmiPyRu, substantially modifies its chemical-physical and biological properties allowing the final formulation a rapid interaction with cell membranes and a quantitative cellular uptake. In this regard, we have already inserted a specific reference of our previous original research paper focusing on the design, development and characterization of PalmiPyRu as the most promising lipophilic derivative of a small set of bioengineered derivatives produced starting from AziRu. Its chemical behaviour and biological properties were investigated in preliminary targeted bioscreens (10.1016/j.bioadv.2022.213016, Bioengineered lipophilic Ru(III) complexes as potential anticancer agents).  Very interesting is the suggestion recommended by the reviewer on the possible interaction of PalmiPyRu with tumor cell lipids. This could lead to a direct alteration of some lipid cellular components by redox processes straightforwardly triggered by ruthenium's ability to shift between different oxidation states. In this regard we have added a small consideration to the discussion on this topic and for this we thank reviewer #2.

4. In TNBC cells, does PalmiPyRu primarily induce ferroptosis, or does it synergize with other RCD pathways? Are there experimental data showing combinatorial effects on cell death pathways?

4. As highlighted in the paper, we believe that through a multimodal mechanism of action different RCD pathways are activated, which in turn can synergize resulting in the overall antitumor effect. This is demonstrated by the use of specific RCD inhibitors (Z-VAD-FMK for apoptosis, chloroquine for autophagy, and Ferrostatin-1 for ferroptosis), but also by confocal microscopy that highlights "mixed" alterations at mitochondrial level. Moreover, the use of selective inhibitors suggests a roughly similar contribution of the main RCD pathways in the induction of the antiproliferative effects of PalmiPyRu. At the moment we are not able to establish what the primary mechanism is, or at least which one goes first. And this is the reason why we have indicated ferroptosis among the mechanisms of RCD triggered by PalmiPyRu. Biomolecular targeting studies should help clarify these aspects in the future.

5. Which TNBC preclinical models were used to validate PalmiPyRu’s activity? What were the key endpoints measured?

5. To validate the biological effects and antiproliferative efficacy of PalmiPyRu, we thoroughly selected a panel of TNBC phenotypes endowed with specific biological features, representing well-established and recognized preclinical in vitro models. Specifically, MDA-MB-231, BT-549 and Hs 578T were selected as TNBC cellular models bearing distinct phenotypic features, i.e., replicative potential and invasive and migratory capacity. All this information, including the source of the cell clones, is extensively reported in the experimental section of the manuscript. The selected endpoints derive from preliminary experiments and from individual cellular responses. Typically, the antiproliferative and cytotoxic biological activities were evaluated in time course experiments reaching endpoints of 72 h by using a range of concentrations of the compounds under examination. Once IC50 values ​​have been established, biological investigations such as analysis of specific cellular responses to treatments in vitro were conducted at endpoints of 48 h to exclude excessive cell death affecting experimental outcomes. HaCaT cells, MCF-10A, Human Hair Follicular Keratinocytes (HHFK) Primary Human Dermal Fibroblast (HDFa) were selected to evaluate cellular responses in human healthy models. This information is also reported in the appropriate sections of the manuscript.

6. What are the key advantages of PalmiPyRu over AziRu in terms of efficacy, stability, or delivery?

6. These aspects related to PalmiPyRu have actually been partially discussed in point 3. Significant differences in cellular responses after treatment in vitro with AziRu or PalmiPyRu are unquestionably due to the conversion of the low molecular weight ruthenium complex into a bioengineered lipophilic derivative, and to very different cellular uptake kinetics (naked AziRu is much more unstable and excessively hydrophilic to easily cross cytoplasmic membranes). Thus, the conversion of AziRu into a lipophilic derivative considerably modifies its chemical-physical and biological properties allowing the final formulation a rapid interaction with cell membranes and a quantitative cellular uptake. This information is introduced in the first part of the paper. Moreover, a specific reference to our previous original research paper focusing on the design, development and characterization of PalmiPyRu is already encompassed in the paper, wherein its chemical behaviour and biological properties are discussed (10.1016/j.bioadv.2022.213016, Bioengineered lipophilic Ru(III) complexes as potential anticancer agents).

7. This is a research article, and the author need not cite an excessive number of references.

7. We have made every effort to limit the number of citations throughout the handwriting of the manuscript, but the complexity of the cellular responses observed during the treatments and the recall to multiple RCD pathways require in our opinion a robust supporting literature. Furthermore, requests from other reviewers for the insertion of additional references do not allow us to narrow down this list.

Round 2

Reviewer 2 Report

Comments and Suggestions for Authors

The author has revised the article in accordance with the comments. This manuscript can be accepted in present form.

Author Response

We thank the reviewer for his positive feedback and for the time he dedicated to us.